# Fair or Unfair: The Moderating Effect of Sustainable CSR Practices on Anticipatory Justice Following Service Failure Recovery

**Yuan-Shuh Lii, May-Ching Ding * and Chih-Huang Lin**

College of Business Department of Marketing, Feng Chia University, 40724 Taichung, Taiwan;
yslii@fcuoa.fcu.edu.tw (Y.-S.L.); linchihh@fcu.edu.tw (C.-H.L.)

**\*** Correspondence: mcding@fcuoa.fcu.edu.tw; Tel.: +886-4-2451-7250

**Abstract:** This paper investigates the relative effect of anticipatory justice on organizational legitimacy and consumer trust that further leads to consumer citizenship behavior following service failure recovery in Taiwan. Further, the moderating role of sustainable corporate social responsibility (CSR) practices is explored. A causal relationship and survey design with a valid sample of 269 respondents was applied. Findings indicated that organizational legitimacy and consumer trust can be restored through anticipatory justice, in particular interpersonal justice and then further mediated consumer citizenship behavior. As a moderator, a high level of sustainable CSR practices had a significantly stronger effect on anticipatory justice and organizational legitimacy than the low level one but only had an effect on interpersonal justice and consumer trust after service recovery. Practical implications are provided for service providers. The value of this research proposes an integrated model with organizational legitimacy and sustainable CSR practice that has not yet been tested in the model of service recovery. In addition, sustainable CSR practice is proposed as a moderator (high and low) that is compared in the level of strength of the relationships. This moderation effect has not been found previously in the process of service recovery.

**Keywords:** sustainable corporate social responsibility; service recovery; anticipatory justice; organizational legitimacy; consumer trust; consumer citizenship behaviors; Taiwan

---

## 1. Introduction

In 2017, the World Bank published world development indicators in which data showed that the service sector accounted for 49.8% of GDP in South Asian, 59.6% in East Asia and Pacific, 64.5% in Europe and Central Asia, and 77% in North America [1]. According to Taiwan's National Statistics, the service industry accounted for 63% of the country GDP in 2017 [2]. Compared to Asian countries, Taiwan's service sector made the highest contribution to GDP in 2017 [1]. Apparently, the service sector is a major economic activity that contributes the most not only to national GDPs globally but also to Taiwan's economy.

Unfortunately, service providers face service failures that are inevitable. Due to the unique intangibility of service characteristics, services cannot be touched or sensed; hence, customer satisfaction is a highly subjective evaluation. This means that not all customers are going to be pleased with a service [3]. If companies cannot completely eliminate service failures, then understanding the process and effectiveness of complaint handling through service recovery can be of considerable value in improving consumer post-recovery satisfaction and retention [4,5]. Indeed, good service recovery can help a company turn a potentially negative situation into a positive one [6]. Previous research has shown that a successful service recovery can have a positive effect on consumer attitudes as well as

behavioral intentions, such as satisfaction, trust, emotions, repurchase intentions, and the spread of positive word of mouth (WOM) [5,7–13]. However, providing successful service recovery remains a challenge for many service organizations [5,14–17].

Organizational legitimacy and corporate social responsibility (CSR) can assist firms in gaining competitive advantages for their well-being and sustainability [18–20]. Organizational legitimacy is rooted in a social contract; organizations cannot exist independently and thus need to build sustainable relationships with society, which can be derived from gaining social power through sustainable CSR practices [21–23]. Our research incorporated this sustainable social concept and makes three contributions to the service failure and recovery literature. First, we examine the relative influence of anticipatory justice facets (distributive justice, procedural justice, interpersonal justice, informational justice) on consumer attitudinal responses. Anticipatory justice has its roots in organizational change, so this paper rationalizes that service failure is an unexpected event (a sudden change in service processes) that service organizations must handle immediately; otherwise, the service failure is likely to affect organizations' legitimacy and long-term survival. In the literature, anticipatory justice can be a key concept in explaining the formation of customer evaluations of organizational responses to a changing event such as service failure [9,10,12,15,24–27]. However, in previous studies, issues of service recovery and their impact on consumers' post-recovery evaluations have focused on investigating the three distinct justice dimensions (distributive, procedural, interactional) as well as the levels of tangible compensation (with/without, or low/medium/high) [12,15,28–30]. Greenberg (1993) proposed that interactional justice consists of two forms of treatment, which might exert different effects on attitudes [31]. Thus, they should be recognized as two distinct dimensions: interpersonal justice and informational justice. Limited research exists concerning interpersonal and informational justice in the services literature, nor does it address specific moderating factors that may affect the effectiveness of service recovery concerning the four anticipatory justice facets [30]. In this regard, we enhance knowledge about service recovery following a service failure by investigating the relative effects of the four anticipatory justice facets on consumers' attitudinal reactions and subsequent behavioral intentions.

Second, we investigate the role of organizational legitimacy in reactions to service failure and recovery. A service failure is an event that might induce consumers to perceive that service providers are irresponsible and make them doubt in the organizations' legitimacy. Legitimacy theory emphasizes that organizations continuously strive to act in ways that are perceived as proper or even admirable in line with social norms, values, and beliefs of the society [18,32]. Therefore, the theory implies a "social contract," meaning that an organization's policies must meet the norms or the expectations of its society [33,34]. The terms of this social contract consist of legal requirements, as well as societal and community expectations [32]. To achieve sustainable development, organizations have to comply with these terms. Thus, a service provider needs to restore its legitimacy in the event of service failure.

Third, we explore the moderating effect of sustainable CSR practices on the relationship between anticipatory justice facets and attitudinal responses following the service failure recovery. CSR is a key component of a firm's marketing communications. By delivering values that meet consumers' expectations, it results in improved corporate performance and reputation, while also helping worthy causes [19,35–38]. Consequently, CSR can be an effective communication tactic and offers shelter or restores corporate legitimacy and consumer trust following service failure [39,40].

## 2. Conceptual Model and Hypothesis Development

### 2.1. The Research Model

Figure 1 illustrates the interactions and relationships of the constructs investigated in the research model. When service failure occurs, consumers are likely to feel anger and betrayal [13], therefore, companies need to respond immediately through anticipatory justice to restore organizational legitimacy and consumer trust toward the company. In addition, CSR theories provide a rationale

for corporations to use sustainable CSR practices as a moderator to influence consumer perceptions of organizational legitimacy and consumer trust that can be recovered after service recovery efforts. As a result, recovered organizational legitimacy and consumer trust then have an effect on consumer citizenship behaviors, which are represented in the model by in-role and extra-role behaviors.

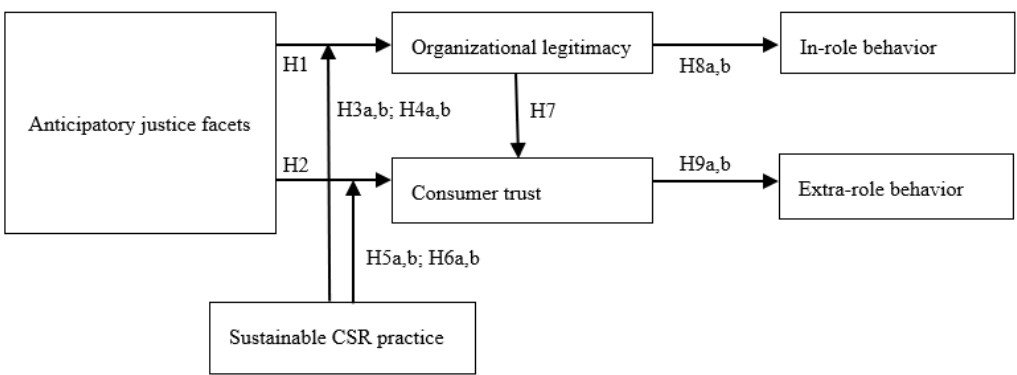

**Figure 1.** The research model.

### 2.2. Service Failure and Recovery Efforts

"Zero defect" is impractical [3]. Even if service providers are machines they may break down sometimes. Service recovery consists of the actions that an organization and its employees take in response to a service failure to restore aggrieved customers to a state of satisfaction [41–43]. From a viewpoint of processes, service recovery involves antecedents, rectified ways to recover from service failure, satisfaction regained after service recovery success, and outcome variables of repurchase intention and WOM when consumers perceive satisfaction after service recovery [44,45].

The concept of justice originated from equity theory [46] that refers to the degree to which consumers feel fairly, rightly, and deservingly treated in comparison to what others have received after the service failure handling and recovery [47–50]. Research on modeling customer satisfaction has primarily focused on one of the roles of expectation as anticipation [51]. Consumers form expectations in line with a level of performance so when service recovery performs well, then satisfaction is high [48,51]. Anticipatory justice has its root in handling organizational change (internal and external environmental change) [52]. That is, anticipatory justice is more case-related and refers to short- and medium-term efforts devoted by the firm. A service failure is an unexpected event that happens internally via employees' (the service providers) wrongdoings during the service processes that, in turn, affect external consumers. Thus, irritated customers should be handled in a way that makes them feel fairly, rightly, and deservingly treated in the process of service recovery.

An effective recovery not only can restore customer satisfaction and trust but also can lead to loyalty (repurchase intention and positive WOM) [53–55]. Studies have shown that ineffective recovery leaves unhappy customers to complain about the service firm to 10 to 20 people [25]. In addition, one study indicated that customers perceive a company's ability in service failure handling or service recovery as the second most important factor in their purchase decision [41]. Furthermore, a successful recovery can prevent customers from spreading negative WOM [29] and influence their purchase intention. These effective service recovery efforts seem to provide sustainable advantage to satisfy or please current customers [45].

### 2.3. The Relative Effect of Anticipatory Justice Facets

Anticipatory justice suggests that the fairness of the recovery procedures, the interpersonal communications and activities, and the outcome are the primary antecedents of customer evaluations in the context of changes (in this case service failure). Traditionally, anticipatory justice facets have been hypothesized identically with four perceived (experienced) forms of justice [52,56], which are a key concept in explaining the formation of customer evaluations of organizational responses to a service

failure. For consumers to perceive fairness in service recovery, service firms take actions (or means) including acknowledging the failure, apologizing, promptly correcting the problem, supplying an explanation for the service failure, empowering staff to resolve problems on the spot, making offers of amends, being courteous and respectful, and showing empathy and attentiveness during the recovery process [7,24,57,58]. Indeed, good service recovery can help a company turn a potentially negative situation into a positive one. Service failure and recovery should be seen as critical decisive moments for firms to gain legitimacy and prove their efforts to satisfy customers and keep them coming back [6].

Distributive justice refers to the perceived fairness and equality that the firm adopts to allocate resources to rectify and compensate for a service failure. It has generally focused on the tangible compensation given to customers during the service recovery, including monetary rewards as refunds, discounts on future purchase, coupons, and exchanging the good or service [7,49,50,59]. Procedural justice refers to the perceived fairness of the specific policies, processes, and methods the firm adopts to handle the service problem and recover from the failed service [7,50], including timeliness, accessibility, process control, and flexibility to adapt to the consumer's recovery needs. Interactional justice refers to the manner in which consumers are treated during the complaint handling process, including elements such as treating consumers with interpersonal sensitivity, courtesy and respect, or providing appropriate explanations for the service failure [7,50,60–62].

Much research on service failure and recovery has taken a three-dimensional approach to experienced (perceived) justice [10,29,50,55]. Greenberg (1993) proposed that interactional justice consists of two forms of treatment, which might exert different effects on attitudes [31]. Hence, these two forms should be recognized as two distinct dimensions: interpersonal justice and informational justice. Interpersonal justice refers to the degree to which people are treated with politeness, respect, and dignity by organizations or employees involved in executing recovery procedures or determining outcomes. Informational justice focuses on whether consumers receive adequate and truthful explanations and justifications for their recovery treatment [26,63]. A four-dimensional view of justice provides a better fit to the measuring model than the three-dimensional approach [63]. Hence, this research adopts the four-dimensional framework of anticipatory justice comprising of distributive, procedural, interpersonal, and informational justice as distinct dimensions.

*2.4. Organizational Legitimacy*

Legitimacy is crucial not only for a firm's existence but also sustainability [18,20,64]. Legitimacy is a perception that "the actions of an entity are desirable, proper, or appropriate within some socially constructed system of norms, values, beliefs, and definitions" [18] (p. 574). Drawing from resource-dependency theory, legitimacy consists of regulative, normative, and cognitive legitimacy [18,65]. Similar to Suchman's tri-level legitimacy theory, organizations established legitimacy at three different levels: pragmatic, moral, and cognitive legitimacy [18,23]. *Pragmatic (regulative) legitimacy* is based on institutional theory that legitimacy is recognized when organizational actions account for the interest of stakeholders in exchange for tangible returns [66]. By addressing stakeholder expectations, organizations gain legitimacy. *Moral (normative) legitimacy* is a conscious level at which organizational behaviors meet the expectation of public judgments of moral and societal norms, and is perceived to be a higher level of legitimacy [23]. The former two provide strategic means for an organization to influence changing events to maintain or repair legitimacy [23]. *Cognitive legitimacy* is based on societal comprehension of an organization's existence and whether its actions are needed or necessary to the society. This taken-for-granted perception is based on social norms [67] and thus is difficult for organization's direct and strategic influence on perceptions manipulation as cognitive legitimacy operates mainly at the subconscious level. Organizations cannot exist independently and thus need to build sustainable relationships with their society. As a result, actions taken by organizations should be in harmony with societal values and expectations [68]. Only when these expectations are met, can organizations continue to operate and survive [69]. Legitimacy is a way that organizations gain cognition and support from their society [70]. In other words,

organizational legitimacy includes tangible and intangible assets and relates to more long-term and stable components of the firm.

Today, rising public perception of organizations' roles in society not only holds a high legitimate standard but also expects them to do things right (for sustainability). Those changes in societal expectations can make it challenging for organizations to operate profitably and remain congruent with societal benefits. This paradox is leading to legitimacy threats that surge when organizations try to compete and survive using wrongful means, resulting in unexpected events such as service failure, corporate scandals, or negative publicity that influence the organization's long-term existence. Legitimacy theory offers a foundation and reason for organizations to perform anticipatory justice in service recovery processes.

### 2.5. Anticipatory Justice and Organizational Legitimacy

As previously mentioned, anticipatory justice tends to be cause-related and refers to short- and medium-term efforts by the firm; organizational legitimacy is related to long-term and more stable assets accumulated by the firm. According to equity theory, consumers experience inequity (unfairness) in the case of service failure. However, consumer perceptions of equality will be restored if anticipatory justice performance is high; meaning that service providers recover appropriately from their failure [10,45]. As such, consumer perceptions of satisfaction following a service failure can be reshaped by their perceptions of equality during the process of service recovery.

Once a service failure occurs, consumers question this organization's legitimacy. This unexpected event (not meeting the regulation or societal norm, value, and beliefs) will impact consumers and break that organization's legitimacy in society temporally. According to equity theory, legitimacy is likely be restored if service recovery via anticipatory justice is performed fairly, rightly, and deservingly to consumers.

Once service failure occurs recovery efforts should start by offering an explanation, fair treatment, effective complaint procedures, and fair compensation [49,50,60,71] to mitigate the impact of service failure and to regain organizational legitimacy. Previous studies have found that the four dimensions of experienced (perceived) justice are significant antecedents of post-recovery satisfaction [5,7,26,72,73]. The effect of the four dimensions of perceived justice on satisfaction with complaint handling is significant [26,54,74] but different across the degree of recovery efforts pertaining to the justice dimensions [7,49,53,55,60,75,76].

Similarly, anticipatory (perceived) justice can provide the rationality and a critical role in repairing perceived legitimacy [21] following service failure recovery. Research showed that anticipatory justice affects individual attitudes and behavior toward changes (events) [52,56]. A service failure that becomes public via press and media coverage is likely to inflict direct damage on organizational legitimacy. Rationally and strategically, there should be an immediate anticipatory (perceived) justice response processes to restore legitimacy. Consequently, the following hypothesis is proposed:

**Hypothesis 1 (H1).** *Anticipatory justice facets (distributive justice, procedural justice, interpersonal justice, and informational justice) have positive and varying effects on organizational legitimacy toward the service failure recovery.*

### 2.6. Consumer Trust

Consumer trust is an important construct in service relationships [77]. The literature on relationships and service marketing has developed many conceptualizations and operationalizations of trust. Sirdeshmukh et al., (2002) conceptualized "consumer trust" as a multi-faceted construct involving front line employee (FLE) behaviors and management policies and practices (MMPs) as distinct facets. Along with the text in their paper, consumer trust might cover behavioral (or conative) and cognitive (or evaluative) dimensions [78]. Consequently, Sirdeshmukh et al., (2002) defined

consumer trust as the expectations held by consumers that the service provider is dependable and can be relied on to deliver on its promises [78]. In other words, consumer trust means a service provider or corporation being reliable, keeping its promises, and not taking advantage of consumers in an exchange relationship. Compared to anticipatory justice, which refers to short- and medium-term efforts accumulated by the firm, consumer trust is related to a long-term and stable attribute of the firm.

Researchers have achieved consensus and believe that trust is a fundamental element in building a firm and durable relationship between consumers and service providers [49,79]. In particular, when consumers buy services before they have tried them, this risk of uncertainty is mitigated by consumer trust [80,81]. In the event of service failure, a successful recovery rebuilds trust because consumers are likely to believe that the service provider or corporation has the honesty and integrity to solve their problems (service failure) [15,78].

In the field study of social justice, customers' trust redevelops when consumers anticipate that performance of service recovery via justice (fair or unfair) has met their expectations. When the performance of anticipatory justice is high, so too is the consumer's satisfaction in service recovery. By contrast, consumers are likely to perceive service providers as untrustworthy when they experience a poor service recovery [82].

Consumer trust is sensitive to how justice is performed when complaints are handled by service providers [81]. Previous research has found that consumers who are satisfied with the service recovery are more likely to exhibit a higher level of trust than are those who are dissatisfied [10,49,53,83,84]. One study found that interactional fairness has a significant direct impact on consumer trust [81]. Further, recovery efforts of distributive, procedural, and interactional fairness are positive and essential for building customer trust in upscale hotels [55] and in E-tail [85]. Consumer trust can be restored when consumers are fairly treated in the recovery of the service failure. The current study extends previous research by testing the explanatory power of the four dimensions of anticipatory justice on consumer trust in firms with respect to service recovery.

**Hypothesis 2 (H2).** *Anticipatory justice facets (distributive justice, procedural justice, interpersonal justice, and informational justice) have positive and varying effects on consumer trust toward the service failure recovery.*

### 2.7. Moderating Effect of Sustainable CSR Practice

Scholars have achieved no real consensus regarding the concept and definition of CSR. The most prevailing CSR definition was probably Carroll's pyramid model, which is based on economic responsibilities topped by legal to ethical and philanthropic responsibilities [86–88]. Recent development has merged the economic, social, and environmental responsibilities to create a strategic approach of creating societal value that also fosters corporate sustainability [89–91].

For sustainable marketing, such unmet social needs provide market opportunities for companies to differentiate and reposition themselves to gain competitive advantages and economic success by fulfilling carefully defined unmet societal needs and benefiting society [19]. As a result, CSR consists of social actions performed by corporations for the purpose of fulfilling social needs [92]. It also involves a corporation's willingness to go beyond its legal obligations to set policies and practices for the benefit of society [93]. In addition, Lerner and Fryxell (1988) stated that a firm's CSR actions should be in harmony with societal values and expectations [94]. Marrewijk (2003) defined CSR as integrating economic, social, and environmental responsibilities in relation to the goal of corporate sustainability and meeting the present needs without sacrificing the needs of future generations [90].

There are many ways of implementing CSR initiatives in marketing practice. For example, there are externality-related and utility-maximizing purposes of CSR practice. To enhance corporate reputation, the philanthropic way of CSR seems to be the most effective way by linking association of brand/cause [95]. The altruistic CSR, defined as the perception of 'giving', might also lower consumers' skepticism toward the firm's intentions, thereby increasing the positive attitude toward

CSR initiatives [96,97] and possibly affecting purchase intentions [98]. A good corporate reputation that might be affected by the three types of CSR initiatives, such as sponsorship, cause-related marketing, and philanthropy, will bring value to the corporation, specifically through consumers' positive attitudes toward the brand and enhanced perceptions of the CSR campaign as credible [99,100]. This perception will affect consumers' extra-role behavior and purchase intention [95].

In line with the literature on service failure response, previous research has explored moderators such as customer emotional responses [101], emotional intelligence [102], relationship quality [73], customer orientation [103], customer attitude toward complaining (ATC) [9], service failure severity [104], and CSR [105]. However, sustainable CSR practice has not yet been examined as a moderator following service failure recovery.

Today, sustainable organizations are built on social performance while meeting economic and legal responsibilities. Sustainable CSR practices gain social power in society that becomes social capital in the long run; then, this capital further secures corporate legitimacy [20]. To win consumer support, corporate behaviors must align with the social norms, values, and beliefs to derive corporate social legitimacy [106]. In a crisis such as service failure, a company must protect and save its social legitimacy and a positive history of CSR offers a possible solution to the crisis [106]. Additionally, studies showed that CSR actions gain legitimacy and support from consumers [21,22,64,107,108]. As a result, this social capital of sustainable CSR is likely to protect organizational legitimacy in an event of service failures.

A positive history of CSR efforts offers a halo effect in protecting corporate brand image and creating credibility and trust [106]. Several studies have concluded that consumer response to CSR efforts can both build and protect corporate brand image, trust, and behavioral intentions [106,109–111] against service failures. Hence, corporations that engage in the process of anticipatory justice to recover corporate legitimacy and consumer trust are likely be moderated by sustainable CSR practice. The following hypotheses are proposed:

**Hypothesis 3 (H3).** *Sustainable CSR practice moderates the effect of anticipatory distributive justice on (a) organizational legitimacy and (b) consumer trust toward the service failure recovery.*

**Hypothesis 4 (H4).** *Sustainable CSR practice moderates the effect of anticipatory procedural justice on (a) organizational legitimacy and (b) consumer trust toward the service failure recovery.*

**Hypothesis 5 (H5).** *Sustainable CSR practice moderates the effect of anticipatory interpersonal justice on (a) organizational legitimacy and (b) consumer trust toward the service failure recovery.*

**Hypothesis 6 (H6).** *Sustainable CSR practice moderates the effect of anticipatory informational justice on (a) organizational legitimacy and (b) consumer trust toward the service failure recovery.*

*2.8. Organizational Legitimacy and Consumer Trust*

A company gains legitimacy through practices and behaviors that have become institutionalized [18,23]. This provides the essence for an organization to stand in the society. Dispositional legitimacy, one form of pragmatic legitimacy, is derived when a company acts in way perceived by consumers as anthropocentric, which in return generates a strong brand image [23] and then this positive image establishes trust in the long run. In addition, moral legitimacy focuses on firms "doing the right thing" and this continuous practice leads to consumer trust. Trust is built when customers have confidence in a service provider's reliability and integrity [79]. As a result, organizational legitimacy strengthens consumers' confidence of the firm's trustworthiness.

Organizational legitimacy is a firm performing an appropriate actual role that aligns with social norms, values, and beliefs. Trust is the confidence that a firm is occupying a specific role and performing

in a manner consistently meeting or exceeding the social expectations (of the actual role) [112,113]. In line with theoretical research, the following hypothesis is presented:

**Hypothesis 7 (H7).** *Organizational legitimacy toward the company has a positive effect on consumer trust toward the service failure recovery.*

*2.9. Consumer Citizenship Behavior*

Consumer-citizens refers to responsible, socially conscious consumers who are willing to make rational decisions and sacrifice self-interests for the common good [114–116]. Consumer tendency or demonstration of citizenship behavior in relation to in-role and extra-role behaviors has its roots in the theory of Organizational Citizenship Behaviors (OCB), which according to Organ (1988), represents individual behaviors that are discretionary and not directly or explicitly recognized by a formal reward system [117]. Simply put, consumer citizenship behaviors can be interpreted or defined as consumer chosen actions that are beneficial to companies in terms of purchases and royal support (in-role behaviors) and spreading good words to relatives and friends (extra-role behaviors).

Social exchange theory supports the link between customer attitudinal evaluations and customer citizenship behavior [118]. The theory explains that relationships between consumers and organizations are seen as social exchanges in which consumers reciprocate a positive gain (e.g., identity experience) from a sense of personal obligation or gratitude by giving positive feedback to the organization. Companies that are perceive as legitimate (aligned with social norms, values, and beliefs) tend to allocate their limited resources to the company [18,23]. In addition, legitimate companies are likely to win positive attitude and support from stakeholders [21]. In one empirical study, consumers supported a community retailer after it gained legitimacy and, in return, benefited the retailer by repurchases and recommendations to friends and relatives (WOM) [22].

Equity theory suggests that low repurchase intention and negative e-WOM could be repaired if firms resolved customers' feelings of inequity following a service failure. Social exchange theory explains that the way we feel about a relationship with another party depends on our perceptions of balance or fairness in a process of reciprocal or negotiated exchanges [118,119]. When consumers interact with a firm during the process of service delivery, the exchange can be viewed as social [120,121]. Social exchange is based on the expectation of trust and reciprocation, as the exact nature of the return is left unspecified [118]. When a service provider establishes its reliability and credibility, consumers perceive the service provider as less risky [82], which has a positive impact on repurchase intention and word-of-mouth [55]. Thus, when consumers are satisfied with the firm's service recovery performance beyond their level of expectation, they are more likely to develop trust in the firm and engage in reciprocal behavior that may benefit the firm [122]. Thus, an effective service recovery can increase consumer trust, which preserves the customer's intent to repurchase the same service from the firm in the future and to engage in positive WOM [9,55,82,123]. Hence, consumers are more likely to engage in reciprocation that may benefit that company [122]. Citizenship behavior may be one type of these benefits.

Based on the concept of OCB, equity, and social exchange theories, consumers are more likely to express their support for legitimate and credible organizations by engaging in in-role behaviors, such as purchasing products from the company, and extra-role behavior, such as making recommendations to others and engaging in positive WOM. Thus, in line with previous research, we propose the following hypotheses:

**Hypothesis 8 (H8).** *Organizational legitimacy has a positive effect on consumer (a) in-role and (b) extra-role behaviors toward the service failure recovery.*

**Hypothesis 9 (H9).** *Consumer trust has a positive effect on consumer (a) in-role and (b) extra-role behaviors toward the service failure recovery.*

### 3. Research Method

*3.1. Design and Sample*

MasterCard conducted a survey on consumer spending behavior in 2016 and found that Taiwan ranks fourth in dining-out spending in the Asia Pacific region behind South Korea, Australia, and Singapore [124]. Since the one thing every service can count on is that there is a service failure in the future [3], this study, similar to the research of Ortiz et al. (2017), investigates service failure recovery during the dining period regarding restaurant services in Taiwan [13].

A self-administered survey was conducted to collect data among Executive MBA (EMBA) students and alumni at a large university in Taiwan. They answered a questionnaire asking them to remember a recent negative experience with a restaurant following the service failure. A total of 450 questionnaires were distributed to EMBA students and 301 stated they had experienced a service failure with restaurant. This study collected 269 valid samples after eliminating 32 with inconsistent responses. Of the 269 participants, 152 (56.5 percent) are male and 117 (43.5 percent) are female. The median age of the participants is 39 years (range 31–63). The majority of the participants are married (87%).

*3.2. Measures*

Multiple-item scales were used to measure each construct. The measurement items were based on previous research, and modified and translated to Chinese to better fit the context of this study. All items were measured on a five-point, Likert-type scale ranging from "strongly disagree" (1) to "strongly agree" (5).

Measurements include anticipatory justice facets, sustainable CSR practices, organizational legitimacy, consumer trust, and consumer citizenship behavior. Measurement of anticipatory justice facets was adapted from Colquitt [63], Homburg and Fürst [59], and Maxham and Netemeyer [61]. For sustainable CSR practice, a measurement was borrowed from Lichtenstein et al. [125] and Berens et al. [126]. Organizational legitimacy items were adopted from Chung, Berger, and DeCoster [66]. Consumer trust was adapted from Crosby et al. [127] with a minor revision to fit this study. Three statements adapted in part from Putrevu and Lord's research were used to measure consumer in-role behavior [128]. Three items adapted from de Matos et al. were used to measure consumer extra-role behavior [9]. The scale and construct items are in Appendix A.

Table 1 provides means, standard deviations, correlations, and scale reliability of constructs. Cronbach's alpha was used to estimate the internal consistency reliability of the scale. These values for subscales ranged from 0.80 (organizational legitimacy) to 0.96 (extra-role behavior). All values surpassed the recommended value of 0.70 [129]. Following Fornell and Larcker, composite reliability (CR) was applied to examine the internal consistency of the multi-item scales included in the model [130]. As shown in Appendix B, the composite reliability of each construct ranged from 0.79 (organizational legitimacy) to 0.97 (consumer extra-role behavior), exceeding the suggested minimum of 0.70 level [131]. The results indicated adequate internal consistency of the measurement model for further analysis of the structural model.

Convergent and discriminant validity tests were also performed to determine construct validity. The standardized factor loadings and average percentage of variance extracted (AVE) were used to measure convergent validity. As suggested by Hair et al., standardized factor loadings with estimates at 0.50 or higher were considered significant [132]. All loadings in the constructs were higher than 0.50 (see Appendix B). In addition, all AVE estimates for each construct were greater than 0.50 (see Appendix B). To assess the discriminant validity, the square root of the AVE in each construct is compared to the correlation coefficients between two constructs [130]. The results showed that the constructs in the measurement model appeared to have acceptable levels of discriminant validity (see Table 1).

**Table 1.** Measurement scales used and properties.

| Constructs | Mean | SD | $\sqrt{AVE}$ | ADJ | APJ | AIFJ | AIPJ | OL | CT | IN | EX |
|---|---|---|---|---|---|---|---|---|---|---|---|
| ADJ | 3.89 | 0.52 | 0.82 | (0.86) | | | | | | | |
| APJ | 3.78 | 0.76 | 0.88 | 0.53 ** | (0.89) | | | | | | |
| AIFJ | 3.83 | 0.57 | 0.79 | 0.50 ** | 0.44 ** | (0.82) | | | | | |
| IPJ | 3.90 | 0.61 | 0.79 | 0.57 ** | 0.55 ** | 0.45 ** | (0.85) | | | | |
| OL | 3.92 | 0.50 | 0.75 | 0.65 ** | 0.58 ** | 0.57 ** | 0.74 ** | (0.80) | | | |
| CT | 4.05 | 0.45 | 0.82 | 0.68 ** | 0.62 ** | 0.58 ** | 0.79 ** | 0.81 ** | (0.85) | | |
| IN | 4.04 | 0.53 | 0.82 | 0.52 ** | 0.47 ** | 0.45 ** | 0.60 ** | 0.69 ** | 0.73 ** | (0.83) | |
| EX | 4.07 | 0.50 | 0.95 | 0.49 ** | 0.45 ** | 0.43 ** | 0.57 ** | 0.65 ** | 0.69 * | 0.87 ** | (0.96) |

Notes: * $p < 0.05$; ** $p < 0.01$; N = 269 and alpha reliability is reported in parentheses in the diagonal. SD: standard deviation, ADJ: anticipatory distributive justice, APJ: anticipatory procedural justice, AIFJ: anticipatory informational justice, AIPJ: anticipatory interpersonal justice, OL: organizational legitimacy, CT: consumer trust, IN: in-role behavior, EX: extra-role behavior.

This study conducted two tests to assess common method variance (CMV). First, this study applied Harman's one-factor test using confirmatory factor analysis by specifying a hypothesized method factor underlying all the manifest variables. As noted by Anderson and Gerbing [133] and Hooper et al. [134], while GFI, AGFI, and NFI were more affected by sample size, the one-factor model presented $\chi^2/df$ (9.98), TLI (0.53), CFI (0.58), PNFI (0.50), and RMSEA (0.18). These values of the model fit were extremely unsatisfactory, indicating that CMV problem was not serious.

Second, in light of possible limitations of Harman's one-factor test, this study employed a marker-variable technique. Lindell and Whitney argued that CMV can be evaluated by identifying a marker variable [135]. According to the testing result, the marker variable was not related to any of the variables in the model. This provided further evidence that CMV was not a problem.

## 4. Data Analysis and Results

### 4.1. Structural Model Analyses

Structural equation modeling (SEM) is usually applied for a causal relationship and can be used to analyze direct and indirect effects of CSR on legitimacy and consumer trust in a single model [136]. Using Amos 21.0 software, the results indicated that the overall model fit showed that the $\chi^2$ statistic ($\chi^2/df$ = 2.97, $p < 0.01$) is unsatisfactory. However, the ratio of $\chi^2$ to degrees of freedom was within the acceptable range for the sample. For the structural model, the TLI (0.90), CFI (0.91), PNFI (0.74), and RMSEA (0.05) values were satisfactory based on the standards suggested by Bagozzi [131].

Following the results of SEM analysis shown in Table 2, the standardized estimates of the model exhibited support for the positive impact of the four facets of anticipatory justice on the organizational legitimacy and consumer trust, thus providing support for H1 and H2. Of them, anticipatory interpersonal justice exerts the strongest effects on organizational legitimacy and consumer trust, followed by anticipatory distributive justice, anticipatory informational justice, and anticipatory procedural justice. Similarly, the result is aligned with one empirical study that found only interactional justice (interpersonal and informational justice) impacted trust [80].

As expected, organizational legitimacy was related in a significantly positive way to consumer trust, supporting H7 and confirming a previous study [113]. Organizational legitimacy significantly affected consumer in-role and extra-role behaviors, supporting H8a and H8b and the result was consistent with a previous study [22]. Further, consumer trust also had a significantly positive effect on consumer in-role and extra-role behaviors, which supported H9a and H9b and confirmed previous studies [9,55,82]. The findings of SEM indicated that consumers who have more positive justice perceptions are more likely to exhibit more positive attitudinal responses such as organizational legitimacy and trust toward the service failure recovery. Consequently, the positive attitudinal responses generated positive in-role behavior (such as encouraging them to continue the service) and extra-role behavior (such as sharing their experiences).

**Table 2.** Results of structural equation modeling (SEM) analysis.

| Path | SR | CR |
|---|---|---|
| Anticipatory distributive justice → Organizational legitimacy | 0.22 | 2.85 ** |
| Anticipatory procedural justice → Organizational legitimacy | 0.13 | 2.01 * |
| Anticipatory informational justice → Organizational legitimacy | 0.20 | 2.87 ** |
| Anticipatory interpersonal justice → Organizational legitimacy | 0.46 | 5.53 ** |
| Anticipatory distributive justice → Consumer trust | 0.15 | 2.22 * |
| Anticipatory procedural justice → Consumer trust | 0.11 | 2.01 * |
| Anticipatory informational justice → Consumer trust | 0.12 | 1.98 * |
| Anticipatory interpersonal justice → Consumer trust | 0.34 | 4.02 ** |
| Organizational legitimacy → Consumer trust | 0.33 | 3.37 ** |
| Organizational legitimacy → In-role behavior | 0.29 | 2.28 * |
| Organizational legitimacy → Extra-role behavior | 0.25 | 2.16 * |
| Consumer trust → In-role behavior | 0.49 | 3.92 ** |
| Consumer trust → Extra-role behavior | 0.49 | 4.30 ** |

Notes: * $p < 0.05$, ** $p < 0.01$; SR: Standardized regression; CR: Critical ratio.

### 4.2. Moderating Effect of Sustainable CSR Practice

A multiple-group SEM analysis was conducted to examine the moderating effect of sustainable CSR practice on the influence of anticipatory justice facets as it relates to organizational legitimacy and consumer trust (H3, H4, H5, and H6). The sample was divided into two groups based on a median split of the sustainable CSR practice score (median = 3.92), which has been a common form of dichotomization in the literature [9,137,138]. The two groups consisted of those consumers who perceived the restaurant to have a lower level of sustainable CSR practices ($n = 108$) and those who perceived a higher level of sustainable CSR practices ($n = 161$). The results were reported in Table 3.

**Table 3.** Moderating effects of sustainable CSR practices.

| Path | $\rho\chi^2$ | Low-Sustainable CSR Practices | | High-Sustainable CSR Practices | |
|---|---|---|---|---|---|
| | | B | t | B | t |
| Anticipatory distributive justice → Legitimacy | 9.12 ** | 0.10 | 0.94 | 0.30 | 2.48 ** |
| Anticipatory procedural justice → Legitimacy | 49.59 ** | 0.20 | 1.63 | 0.17 | 2.01 * |
| Anticipatory interpersonal justice → Legitimacy | 12.36 ** | 0.21 | 1.49 | 0.36 | 3.45 ** |
| Anticipatory informational justice → Legitimacy | 26.42 ** | 0.24 | 2.53 * | 0.32 | 2.85 ** |
| Anticipatory distributive justice → Trust | 9.30 ** | −0.04 | −0.51 | 0.19 | 1.60 |
| Anticipatory procedural justice → Trust | 73.62 ** | −0.04 | −0.65 | 0.13 | 1.41 |
| Anticipatory interpersonal justice → Trust | 11.32 ** | 0.53 | 3.62 ** | 0.75 | 8.46 ** |
| Anticipatory informational justice → trust | 12.20 ** | 0.08 | 1.21 | 0.18 | 1.43 |

Notes: * $p < 0.05$, ** $p < 0.01$.

Considering the moderating effect of sustainable CSR practices on the relationship between the four anticipatory justice facets and organizational legitimacy, the results showed that the $\chi^2$ difference tests for both unrestricted and restricted models were statistically significant. Similarly, the moderating effect of sustainable CSR practices was also found in the relationship between anticipatory justice and consumer trust. Further, the standardized path coefficients were all higher in the high-sustainable CSR practice group than in the low-sustainable CSR practice group, indicating that the effects of the four anticipatory justice facets on organizational legitimacy were greater in the high-sustainable CSR group than in the low-sustainable CSR group. The results supported H3a, H4a, H5a, and H6a.

However, the standardized path coefficients of the moderating effect of CSR sustainable practice was only found in the relationship between anticipatory interpersonal justice and consumer trust, which supported H5b but not H3b, H4b, and H6b. The positive coefficient showed that the beneficial effect of anticipatory interpersonal justice on consumer trust was significantly stronger

in the high-sustainable CSR group ($B = 0.75$, $p < 0.01$) than in the low-sustainable CSR group ($B = 0.53$, $p < 0.01$), however, no matter how high or low the sustainable CSR practice, it can significantly moderate interpersonal justice and consumer trust.

## 5. Discussion and Conclusions

### 5.1. Discussion of Empirical Findings

This study had two objectives. The first was to examine whether corporate responses (anticipatory justice) can help companies restore organizational legitimacy and consumer trust as well as mediated consumer citizenship behaviors following service failure recovery. All hypotheses proposed by the structural model were statistically significant. The second was to investigate the moderating role of sustainable CSR practices on anticipatory justice to influence the recovery of organizational legitimacy and consumer trust after the process of service recovery. The results were mixed. The moderating role between low- and high-level of sustainable CSR practices was significant between all four facets of anticipatory justice and organizational legitimacy and consumer trust. In addition, the standardized path coefficients were all significant between the relationship of anticipatory justice and legitimacy for the high level of sustainable CSR practice group but only interpersonal justice and consumer trust were significant for both high and low level of sustainable CSR practices. This may mean that as long as companies put efforts into sustainable CSR practices (no matter high or low) that consumers honor the company for doing the right thing. The results in this study contribute to the service recovery literature and have practical implications for the service sector.

This study's academic contribution is to construct a research model based on an integration of equity, legitimacy, OCB, and CSR theories in service failure and recovery literature. Additionally, legitimacy theory is essentially a social contract between organizations and society that offers a foundation and justification for corporations to take recovery actions (processes of anticipatory justice). As suggested by organizational legitimacy theory, companies cannot operate independently; they have to act in line with social norms and expectations for sustainability. To consolidate legitimacy, companies need to foster sustainable CSR practices that gain social power (social capital) to rectify problems such as service failure. As a result, companies are attempting to justify their continued legitimated existence and leveraging sustainable CSR practices to demonstrate good corporate citizenship, which then leads to consumer citizenship behaviors.

For practical implications, anticipatory justice did exert significant and different relative effects on organizational legitimacy. As a result, service providers who want to recover organizational legitimacy and consumer trust after service failure should, first and foremost, craft interpersonal skills followed by distributive, informational, and procedural justices. In handling service failure, companies should first provide an emotional training program to their service providers so they can immediately respond in an interpersonal manner, with sincere apologies, politeness, and expressions of sympathy that let consumers feel that service providers are on their side. This is likely to quickly calm the irritated consumers. Second, a compensation program gives service providers a certain power to offer compensations in monetary and non-monetary ways as a tool to handle service failure. After irritated consumers calm down, this is a good time to offer appropriate compensations. Third, service providers should honestly explain to the customer why the service failed. Last, fair procedure policies and practices should be implemented in a way that consumers see as clear, consistent, fair, and moral in handling the problem. After those recovery processes are taken, consumers are likely to perceive the recovery as fair, responsible, and reliable. As a result, emotionally imbalanced consumers are likely to be pleased and the service providers may regain organizational legitimacy and consumer trust.

According to Suchman, pragmatic and moral legitimacy provide strategic means for an organization to manipulate events to gain or repair legitimacy [18]. As pragmatic legitimacy is based on social exchange theory, it exists when organizational practices are in line with societal values and expectations [68] and moral legitimacy is a higher order of legitimacy in that sustainable

organizations act in a moral way (do the right thing). As a result, consumer-citizens—responsible and socially conscious consumers—are willing to support sustainable organizations by consumer citizenship behaviors (repurchase and positive WOM) reciprocally. Thus, the practical implication for service providers is to pursue moral legitimacy, setting it up as a company mission or policy such as a code of conduct to foster "doing the right thing" that is congruent with consumer and societal expectations. Remember, a reward system is essential to promote the right behavior in the long run, such that eventually everyone will act in legitimate and moral ways.

Similarly, consumer trust is rebuilt as long as consumers perceive that the service providers are responsible, honest, and keep promises in solving service problems. When organizations set up practices that follow moral legitimacy, trust between the organization and consumers develops into a durable service relationship. This trust, reinforced by organizational legitimacy, is reliable and durable so that consumers reward the company by in-role and extra-role citizenship behaviors. The foremost policy to be followed by companies is that service providers must never lie to customers.

The moderating role of sustainable CSR practice is apparent in that a high level of sustainable CSR practices have a significant impact between anticipatory justice and organizational legitimacy and consumer trust after service recovery. The results point to practical implications: companies should exercise a high level of sustainable CSR practice, which can have a halo effect in restoring organizational legitimacy and consumer trust following service failure recovery. Service providers can implement a company policy or practice for the benefit of society. In particular, sustainable companies can give back to society (or specific communities) by donations, taking a portion of corporate profitability every year. Consumers are unlikely to be skeptical of companies that practice philanthropic CSR, an altruistic type of CSR [97]. This giving back to society has to be consistent over time. In the long run, company reputation of high level of sustainable CSR practices will imprint in the mind of consumers. Once this reputation is built, the company can leverage its halo effect to regain organizational legitimacy and in particular consumer trust after service recovery. To date, consumers are conscious and demand corporations to act and meet societal values. Reciprocally, consumers are willing to pay extra to support (repurchase and positive WOM) such sustainable companies.

Furthermore, marketing communication of sustainable CSR practices should be in place but executed in a delicate way. Service providers should promote good deeds with a purpose to have more people join in to do the right thing (via social network). Overall, consumers tend to believe a company with sustainable CSR practice that leads to corporate legitimacy and consumer trust, which then leads to consumer citizenship behavior. Today, sustainability is established on social performance; thus, sustainable CSR practices gain social power and this social capital further secures or restores organizational legitimacy and consumer trust if service failure occurred.

*5.2. Concluding Remarks*

A limitation of this research is in its sample and restaurant context, which constrain the generalizability (external validity) of this study. Future studies can use a random sample and examine a greater breadth of service industries. Further investigation of different types of organizational legitimacy, in particular the moral legitimacy, should be considered. In addition, moral legitimacy interaction with sustainable CSR practices as a social capital provides another direction for future research in the process of service recovery. A dichotomization method was used to split responses of sustainable CSR practice into high and low levels. Different scenarios (experimental design) of CSR practices can be considered and examined in the future. Moreover, the relationship between organizational legitimacy and consumer trust is significant in this study but warrants further investigation to support this result. In subsequent research, different types of legitimacy in relation to consumer trust may be an interesting topic that adds to the field of service recovery.

**Author Contributions:** All authors contributed equally in the study's model, research method, and the results. However, the following revision of comments was handled by correspondence but all authors have reviewed, commented, and approved the final manuscript.

**Funding:** This research received no external funding.

**Conflicts of Interest:** We declare no conflict of interest.

## Appendix A. Measurement Construct Items

**Table A1.** Measurement scales and construct items.

| Distributive justice (Colquitt, 2001; Homburg and Furst, 2005; Maxham and Netemeyer, 2003) [59,61,63] |
| --- |
| • Gives the inconvenience caused by the problem and the time lost, the compensation I received from the restaurant has been correct. |
| • I think the restaurant has been fair when compensating me for the problem that occurred. |
| • In general, the outcome I received from the restaurant in response to the problem in the service performance has been adequate. |
| Procedural justice (Colquitt, 2001; Homburg and Furst, 2005; Maxham and Netemeyer, 2003) [59,61,63] |
| • I think the restaurant has fair policies and practices for dealing with problems. |
| • The restaurant has shown flexibility in solving the problem. |
| • The restaurant tried to solve the problem as fast as possible. |
| Interpersonal justice (Colquitt, 2001; Homburg and Furst, 2005; Maxham and Netemeyer, 2003) [59,61,63] |
| • In response to the problem, the employee in this restaurant has been honest. |
| • The employee in this restaurant has treated me with courtesy when solving the problem |
| • The care and communication with the restaurant employees to solve the problem have been appropriate. |
| Informational justice (Colquitt, 2001; Homburg and Furst, 2005; Maxham and Netemeyer, 2003) [59,61,63] |
| • The restaurant is candid in communications with me. |
| • The restaurant explains thoroughly the procedures used to make decisions about my complaint. |
| • The restaurant communicates details in a timely manner. |
| Organizational legitimacy (Chung et al., 2016) [66] |
| • I think the restaurant is a necessary part of our society. |
| • I think the restaurant is honest. |
| • I believe the restaurant follows government regulations. |
| Consumer trust (Crosby et al., 1990) [127] |
| • I believe the restaurant can be relied upon to keep its promise |
| • I believe the restaurant is trustworthy. |
| • I would find it necessary to be cautious in dealing with this restaurant. (R) |
| Sustainable CSR practice (Lichtenstein et al., 2004; Berens et al., 2005) [125,126] |
| • The restaurant's business practices are better than the industry's code of conducts. |
| • The restaurant is already committed to using a substantial portion of its profits to help community groups. |
| • The restaurant's reputation for socially responsible behavior is above average for the industry. |
| Consumer in-role behavior (Putrevu and Lord, 1994) [128] |
| • I consider this restaurant as one of my choices in the area. |
| • This restaurant is on my list when making a purchase decision on this type of service. |
| • I will keep going to the restaurant. |
| Consumer extra-role behavior (De Matos et al., 2009) [9] |
| • I will tell my relatives and friends about the good deed of the restaurant. |
| • I will join activity undertaken by the restaurant in the future. |
| • I will recommend the restaurant to my relatives and friends. |

## Appendix B. Measurement Model

**Table A2.** Measurement model and factor loading.

| Maximum Likelihood Estimates | | | | | |
|---|---|---|---|---|---|
| **Constructs** | **Factor Loading** | **Measurement Error** | **SMC** | **CR** | **AVE** |
| Distributive justice (DJ) | | | | 0.86 | 0.67 |
| DJ1 | 0.80 | 0.36 | 0.63 | | |
| DJ2 | 0.87 | 0.25 | 0.75 | | |
| DJ3 | 0.80 | 0.37 | 0.64 | | |
| Procedural justice (PJ) | | | | 0.91 | 0.77 |
| PJ1 | 0.97 | 0.05 | 0.93 | | |
| PJ2 | 0.65 | 0.58 | 0.42 | | |
| PJ3 | 0.98 | 0.07 | 0.95 | | |
| Interpersonal justice (IPJ) | | | | 0.83 | 0.62 |
| IPJ1 | 0.79 | 0.28 | 0.62 | | |
| IPJ2 | 0.76 | 0.42 | 0.58 | | |
| IPJ3 | 0.85 | 0.38 | 0.72 | | |
| Informational justice (IFJ) | | | | 0.83 | 0.62 |
| IFJ1 | 0.78 | 0.48 | 0.61 | | |
| IFJ2 | 0.85 | 0.28 | 0.72 | | |
| IFJ3 | 0.72 | 0.39 | 0.52 | | |
| Organizational legitimacy (OL) | | | | 0.79 | 0.56 |
| OL1 | 0.69 | 0.53 | 0.47 | | |
| OL2 | 0.73 | 0.47 | 0.53 | | |
| OL3 | 0.82 | 0.32 | 0.68 | | |
| Consumer trust (CT) | | | | 0.86 | 0.67 |
| CT1 | 0.73 | 0.47 | 0.53 | | |
| CT2 | 0.84 | 0.30 | 0.70 | | |
| CT3 | 0.88 | 0.23 | 0.77 | | |
| Consumer in-role behavior (IN) | | | | 0.83 | 0.67 |
| IN1 | 0.83 | 0.32 | 0.86 | | |
| IN2 | 0.73 | 0.47 | 0.53 | | |
| IN3 | 0.79 | 0.38 | 0.62 | | |
| Consumer extra-role behavior (EX) | | | | 0.97 | 0.90 |
| EX1 | 0.94 | 0.11 | 0.89 | | |
| EX2 | 0.96 | 0.08 | 0.92 | | |
| EX3 | 0.95 | 0.10 | 0.90 | | |

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
