# Peer review of "Fair or Unfair: The Moderating Effect of Sustainable CSR Practices on Anticipatory Justice Following Service Failure Recovery"

_sustainability, doi:10.3390/su10124548_

Round 1
Reviewer 1 Report
I have some comments to the authors of this article attached to this web-page.

Author Response
Really grateful for your help. Great comments really help to uplift the paper.
Thank you so much that we have learnt from you.

Reviewer 2 Report
Anticipatory justice is an interesting concept, and this paper seeks to show its relevance in business practice.
Underlying the research is the idea that perfection is unattainable. I agree with that, from both a practical and a conceptual basis. However the repeated claim in the paper (p1 ln39; p3 ln125; p10 ln346) needs to be given some grounding.
The study is Taiwan-based and information provided does place that in context. I suggest that the location of the study be mentioned in the abstract and keywords.
Author Response
Thank you so much for your help. Really grateful.
Best regards,
May-Ching Ding

Reviewer 3 Report
The manuscript entitled “Fair or unfair: The moderating effect of sustainable CSR practices on anticipatory justice following service failure recovery” (Manuscript ID: sustainability-380918) investigates the relative effect of anticipatory justice on organizational legitimacy and consumer trust, as well as the the moderating role of sustainable CSR practices. The paper is well documented and properly organized. Hence, the study is worth publishing, but after several revisions, as detailed below.
The introductory section should argue why the framework of China was selected in order to develop the analysis. Therewith, I suggest the author(s) to provide an outline of the entire paper.
The second section discusses in a suitable way previous research. However, this section should be appended with the following studies:
Angus Duff - Corporate social responsibility as a legitimacy maintenance strategy in the professional accountancy firm, The British Accounting Review, Volume 49, Issue 6, November 2017, Pages 513-531
https://doi.org/10.1016/j.bar.2017.08.001
Philipp Bachmann, Diana Ingenhoff - Legitimacy through CSR disclosures? The advantage outweighs the disadvantages, Public Relations Review, Volume 42, Issue 3, September 2016, Pages 386-394
https://doi.org/10.1016/j.pubrev.2016.02.008
Qinqin Zheng, Yadong Luo, Vladislav Maksimov - Achieving legitimacy through corporate social responsibility: The case of emerging economy firms, Journal of World Business, Volume 50, Issue 3, July 2015, Pages 389-403
https://doi.org/10.1016/j.jwb.2014.05.001
Ştefan Cristian Gherghina, Liliana Nicoleta Simionescu - Does Entrepreneurship and Corporate Social Responsibility Act as Catalyst towards Firm Performance and Brand Value? International Journal of Economics and Finance, Volume 7, Issue 4, April 2015, Pages 23-34
http://dx.doi.org/10.5539/ijef.v7n4p23
The third section reveals the empirical methodology. In as much as the author(s) employ the structural equation modeling (SEM) approach, I suggest that this method should be discussed. As well, references to previous studies that applied SEM analysis are necessary.
The fourth section is dedicated to empirical findings. The quantitative setting would be improved if the author(s) employ a data mining technique such as factor analysis.
The last section discusses the quantitative outcomes and delivers concluding remarks. I recommend the author(s) to divide this section into “Discussion of empirical findings” and “Concluding remarks”.
Regarding the heretofore-stated comments, I mention that major revisions are required before reassessing the manuscript.
Author Response
Thank you for your valuable comments. We added some references as you suggested and made other changes. We believe that these changes have lifted the paper.
Best regards,
May-Ching Ding

Round 2
Reviewer 3 Report
The author(s) incorporated most of the suggested changes in the revised version of the manuscript. Hence, the paper improved considerably. However, the study would improve further if a discussion towards entrepreneurship and corporate social responsibility will be undertaken. In this sense, I recommend the author(s) the following paper, also suggested in the first review round:
Ştefan Cristian Gherghina, Liliana Nicoleta Simionescu - Does Entrepreneurship and Corporate Social Responsibility Act as Catalyst towards Firm Performance and Brand Value? International Journal of Economics and Finance, Volume 7, Issue 4, April 2015, Pages 23-34
http://dx.doi.org/10.5539/ijef.v7n4p23
After these modifications, the paper should be published.